# Defining the Minimal Long-Term Follow-Up Data Elements for Newborn Screening

**DOI:** 10.3390/ijns10020037

**Published:** 2024-05-15

**Authors:** Yvonne Kellar-Guenther, Lauren Barringer, Katherine Raboin, Ginger Nichols, Kathy Y. F. Chou, Kathy Nguyen, Amy R. Burke, Sandy Fawbush, Joyal B. Meyer, Morna Dorsey, Amy Brower, Kee Chan, Mei Lietsch, Jennifer Taylor, Michele Caggana, Marci K. Sontag

**Affiliations:** 1Center for Public Health Innovation, Evergreen, CO 80439, USA; marci.sontag@cphinnovation.org; 2Childrens Hospital Colorado, Aurora, CO 80045, USA; lauren.barringer@childrenscolorado.org; 3Connecticut Newborn Screening Network, Connecticut Children’s, Hartford, CT 06106, USA; kraboin@connecticutchildrens.org (K.R.); gnichols01@connecticutchildrens.org (G.N.); 4Newborn Screening Program, Wadsworth Center, New York State Department of Health, Albany, NY 12208, USA; kathy.chou@health.ny.gov (K.Y.F.C.); michele.caggana@health.ny.gov (M.C.); 5Division of Allergy & Immunology, Department of Pediatrics, University of California, San Francisco, CA 94143, USA; kathy.nguyen3@ucsf.edu (K.N.); morna.dorsey@ucsf.edu (M.D.); 6North Dakota Health & Human Services Newborn Screening Program, Bismarck, ND 58505, USA; arburke@nd.gov (A.R.B.); jbmeyer@nd.gov (J.B.M.); 7HealthTech Solutions, Frankfort, KY 40601, USA; sandy.fawbush@healthtechsolutions.com; 8American College of Genetics and Genomics, Bethesda, MD 20814, USA; abrower@acmg.net (A.B.);

**Keywords:** newborn screening, public health, equity, long-term follow-up data

## Abstract

Newborn screening (NBS) is hailed as a public health success, but little is known about the long-term outcomes following a positive newborn screen. There has been difficulty gathering long-term follow-up (LTFU) data consistently, reliably, and with minimal effort. Six programs developed and tested a core set of minimal LTFU data elements. After an iterative data collection process and the development of a data collection tool, the group agreed on the minimal LTFU data elements. The denominator captured all infants with an NBS diagnosis, accounting for children who moved or died prior to the follow-up year. They also agreed on three LTFU outcomes: if the child was still alive, had contact with a specialist, and received appropriate care specific to their diagnosis within the year. The six programs representing NBS public health programs, clinical providers, and research programs provided data across multiple NBS disorders. In 2022, 83.8% (563/672) of the children identified by the LTFU programs were alive and living in the jurisdiction; of those, 92.0% (518/563) saw a specialist, and 87.7% (494/563) received appropriate care. The core LTFU data elements can be applied as a foundation to address the impact of early diagnosis by NBS within and across jurisdictions.

## 1. Introduction

Infants born in the United States (US) are universally offered newborn screening (NBS) for specific medical conditions shortly after birth [1,2]. Ultimately, the goal of NBS is to improve the quality of life for individuals as a result of early detection and treatment commencement [1]. The Centers for Disease Control and Prevention (CDC) has identified NBS as one of the most significant public health achievements of the 21st century [3]; however, the efficacy of this program has not yet been tested [1].

Since 2006, it has been argued that long-term follow-up (LTFU) is needed for NBS to be a meaningful public health activity [4]. Historically, LTFU has been defined as starting once a child has a confirmed diagnosis as a result of an abnormal newborn screen. The length of NBS LTFU has been proposed to vary from diagnosis up until school age, to 18 years of life, or throughout the entire lifespan [5,6]. The ownership and responsibility of managing a LTFU program is not defined and could be housed with the NBS public health program, pediatric clinical providers, or clinical specialists. Currently, the success of NBS in the US is measured using quality indicators that focus on process measures [1]. The US NBS field has had a difficult time, however, gathering data reliably and consistently to determine the long-term impact of NBS on the quality of life for children/adults who were identified with a condition through newborn screening and determining if they are connected to and receiving care for their condition. Identifying and defining minimal LTFU data elements as a starting point and then creating a system designed to collect LTFU data could provide the insight needed to improve and refine the NBS system.

### 1.1. Measuring LTFU for Children Diagnosed with an NBS Disorder

There have been many publications on what LTFU metrics should be used to address the question of the impact of NBS. These include: (1) whether the child is still alive [6,7]; (2) healthcare utilization for both specialty care and primary care (e.g., linked to care; receiving appropriate clinical monitoring and treatment for condition) [4,5,6,7,8,9]; (3) child’s health status (e.g., growth, development, function) [7]; (4) quality of life for the child and family [7]; and (5) ensuring all families obtain care/no disparities in LTFU [7].

While there have been a few research projects that have gathered and looked at LTFU data elements [4,9] and one tool, the Longitudinal Pediatric Data Resource (LPDR) that captures genomic and phenotypic data over the lifespan of NBS-identified newborns assessing the impact of early detection and treatment [10], there has been difficulty gathering LTFU data in the US for children with a newborn screen disorder consistently and reliably—from public health programs or clinical care [4]—to understand the impact of NBS and whether families are receiving needed care. There is also a lack of standards for data elements, sources, and case definitions with regard to LTFU [4,6,8]. The lack of standards may be because there are different stakeholder groups who can provide insight into the effectiveness of LTFU. NBS is a complex system [2] that involves public health programs, hospitals, birthing providers, couriers, families, insurers, and healthcare providers. This system is intended to be comprehensive, encompassing screening, diagnosis, and long-term care for children with a condition identified through an abnormal newborn screen [11]. These groups can provide data from sources to which they have access and bring their perspectives of what successful LTFU looks like. While stakeholders can have different ideas on what to collect for LTFU or have different abilities to collect data, these systems may converge around a few key LTFU indicators.

### 1.2. Minimal LTFU Data Elements

There has been difficulty gathering LTFU data reliably and consistently from the majority of US NBS systems because of the variety of data elements suggested and tested for in LTFU. While this variety mirrors the complexity of LTFU, the authors of this paper feel that it may be more important to find a starting place that allows more states to contribute data and then work towards gathering the more complex data elements. As Lloyd-Puryear and Brower [2] recommended, we are attempting to start incrementally, allowing us to create concrete data definitions. Thus, we are proposing the NBS LTFU minimal data elements. The goal is to identify the bare minimum data elements of interest to all LTFU stakeholders that can be gathered reliably, consistently, and with minimal burden to the existing NBS infrastructures. We recognize that these data will generate more questions. Still, it is beginning to answer whether affected children reap the intended benefits of NBS for improving long-term health by first quantifying how many receive appropriate care over time.

## 2. Materials and Methods

Six awardees—four programs directly managed by or closely associated with state NBS systems (CT, CO/WY, NY, ND), one university program (UCSF), and one professional foundation (the American College of Medical Genetics (ACMG))—were funded by HRSA (HRSA-21-079) to “expand the ability of state public health agencies to provide screening, counseling and services” to the families of newborns and children diagnosed with a condition as a result of an abnormal newborn screen. “The purpose of the program is to support comprehensive models of long-term follow-up that demonstrate collaborations between clinicians, public health agencies, and families” (HRSA-21-079 funding opportunity announcement, pg. (i). Each awardee was provided with the ability to define the methods to create the models for long-term follow-up programs as well as the scope of the programs in terms of disorders included and years of follow-up data collected (see Table 1).

During a regularly scheduled meeting between all six programs, there was a discussion on working towards core LTFU data elements. As part of these discussions, the six programs talked about the different stakeholders who would use LTFU data (see Figure 1) and what questions each stakeholder group might answer from LTFU data. As part of this discussion, a diagram was created (see Figure 1) and goals around creating the core LTFU data elements were created.

A smaller group then met to identify the core LTFU data elements that the six programs could test. To start, the six programs compared the data their programs were already gathering, how it was gathered (e.g., source), and the data format (e.g., drop-down list, yes/no, numeric value). Because of the lack of consistency between the programs’ LTFU data, the decision was made to focus on the core data elements proposed in 2016 by a workgroup of NBS professionals organized by the National Coordinating Center (NCC) of the Regional Genetics Network and the Newborn Screening Translational Research Network (NBSTRN) [12] (see blue box in Figure 1) and test whether the six programs could gather these minimal LTFU data elements to address the effectiveness of NBS.

### 2.1. Approach

The six programs met on 9–10 May 2023; five attended in person, and one program attended virtually. The goal of the meeting was to identify the minimum needed data elements that the majority of awardees could reliably and consistently collect with minimal burden. The participants created a list of data elements that were collected by previous programs as well as their own as minimal LTFU data elements: (1) diagnosis, (2) if the child was still alive, (3) date of appropriate first intervention, and (4) if the child received care and treatment within the last 12 months specific to the diagnosis, and if yes, (5) the type of care provider the child saw. As part of this discussion, the group looked at each LTFU data element proposed and discussed (1) the data element, (2) potential data sources, (3) if the element could be obtained from all participating awardee programs reliably and efficiently, and (4) what were the lowest common data values that could be gathered. Next, the group discussed what was needed to create a common denominator so that percentages could be generated.

The group agreed to pull 2022 data, meet again to discuss the process, and clarify definitions and the data elements as needed. For the data pull, children were grouped based on age on the last day of the evaluation year, 31 December 2022. To measure the reach of LTFU and protect individual privacy, the group gathered long-term follow-up data from 2022 on groups of children in the same birth cohort. Specifically, we grouped children born in 2018 (4 years to <5 years), 2019 (3 years to <4 years), 2020 (2 years to <3 years), 2021 (1 year to <2 years), and 2022 (<1 year) and reported on their status in 2022 for the five questions mentioned above. Programs were asked to report on data within the scope of their follow-up programs to determine if the data elements could provide a structure for LTFU across various programs.

### 2.2. LTFU Data Collection

To date (February 2024), the group has pulled data three times to test and refine the process. All six programs have developed systems to extract data. Table 1 shows the sources used to pull data and the limitations noted by each program.

## 3. Results

### 3.1. Defining Data Elements

Nine initial minimal LTFU data elements were evaluated. Two data elements were included to create a denominator: (1) diagnosis and (2) if the child was deceased or had moved out of the jurisdiction. Another five data elements were included as a potential outcome of NBS success: if the child was still alive, age of first documented contact per the NewSTEPs’ definition of medical intervention by disorder [13], if the child received appropriate care specific to the diagnosis within the last 12 months and is receiving LTFU care, number of children lost to follow up, the number of children actively engaged (for opt-in programs), and if a primary care provider or specialist is seeing the child. Finally, two descriptive minimal LTFU data elements were included in hopes they could provide insight into the outcomes–data about the cause of death, if applicable, and the type of care provider who provided care and treatment to the child (PCP, specialist, or both).

It quickly became clear that gathering all nine data elements was difficult for a few reasons. First, because most programs relied on EHRs or data from specialty clinics, it was not possible to know if every child had seen a PCP. It was also difficult for some of the programs to access the NewSTEPs data, and a few who could access these data noted it was problematic to resolve missing data because the NewSTEPs data does not have identifiers. The group also found that it was difficult to determine if the cause of death was related to the genetic condition because of the many ways the cause of death can be recorded. The group agreed to two data elements to create a denominator and three data elements to measure LTFU outcomes that were potential indicators of NBS success (see Table 2).

#### 3.1.1. Denominator

For the minimal LTFU data elements, the denominator represents the number of children within the birth cohort who have been diagnosed as having a condition that was screened for using NBS dried blood spots, including those cases who were diagnosed after a false negative NBS, minus those who died or moved their care out of the jurisdiction before the year being reported (i.e., in our case, those who died or moved before 1 January 2022).

Diagnosis was a descriptive variable to pull, report, and analyze. It was included for two reasons: (1) the data may come from different clinics, and (2) diagnosis informs what healthcare the child should receive (appropriate care specific to the diagnosis) and allows for data users to see if there is a difference in LTFU outcomes by disorder. For this project, each of the six programs was able to define groups of disorders differently for the purposes of their LTFU systems. The intent of this work is to demonstrate the use of the tool to collect and summarize data, not to report on the state of LTFU by disorder or infer differences in LTFU patterns between programs or disorders. The group categorized NBS disorders as follows: (1) metabolic conditions; (2) congenital adrenal hyperplasia (CAH); (3) CH; (4) hemoglobinopathies; (5) CF; (6) SCID; (7) T-cell lymphopenia; (8) SMA; and (9) X-ALD. The metabolic conditions were categorized together due to similar follow-up recommendations and clinical specialists; the same was performed for the hemoglobinopathies, which included the three core RUSP conditions (S, βeta-thalassemia, S,C disease, S,S disease). T-cell lymphopenias were tracked by one program as a secondary RUSP condition that required comprehensive LTFU. A child was determined to fit into the diagnosis category if the program providing information confirmed the child had that diagnosis using the established NBS public health case definitions [14].

#### 3.1.2. Numerators

There were three final, minimal LTFU data elements designed to capture the potential benefits of NBS. These were (1) if the child was still alive, (2) if the child had contact with a specialist for their disorder within the last 12 months, and (3) if the child received appropriate care specific to the diagnosis within the last 12 months.

##### Alive

One goal of newborn screening is to save the child’s life. As a result, a key LTFU minimal data element is whether the child is alive or not. A child was counted as alive if they were not classified as deceased in the clinic’s electronic health record (EHR), state vital record system, or reported as deceased to the LTFU program.

##### Received Appropriate Care and Treatment Specific to Diagnosis

Another goal of NBS, specifically ST and LTFU, is connection to treatment for children with a diagnosed condition. The group discussed that only some conditions require seeing a specialist annually. As a result, the minimal data elements should capture the number of cases receiving appropriate care for their condition based on recommendations within a 12-month period. This was altered during discussion to answer two questions: (1) did the child have at least one contact with a specialist either in-person, via telehealth, email, or a phone call, and (2) did the child see the appropriate specialist on the recommended cadence for care within the state/jurisdiction (e.g., quarterly visits for children with cystic fibrosis). Appropriate care was defined as the number of children seeing the appropriate specialist on the recommended cadence for care within their state/jurisdiction.

### 3.2. Analysis of LTFU Data

Long-term follow-up data for 672 children were submitted by the six programs for 2022. Five birth cohorts were requested (2018–2022); due to the structure of each program, not all programs could report for all disorders or all birth cohorts (see Table 3). The number of children identified through newborn screening for each group of disorders was provided. Some children moved their care out of the jurisdiction or died before 2022, leaving 563 eligible for LTFU at the beginning of the reporting year (2022). Of those 563, over 92.0% had at least one documented contact with an appropriate care provider; this proportion decreased slightly in older age cohorts (see Table 3). Across all disorders, this number dropped from 95.3% for the 2022 cohort to 85.1% for the 2018 cohort. This contact may have been in the form of a clinic visit, a telehealth visit, or a phone call; these visits were documented to confirm that the specialist was still in contact with the child.

Programs also reported the number of children who met the program-defined condition-specific recommendations for care. While fewer children met those guidelines, 87.7% did in aggregate, and >82.1% met the guidelines across all age groups when looking at all NBS disorders.

Disorder-specific numbers are presented for each cohort, but no comparisons can be made due to small numbers and differences between the disorders reported by programs.

Follow-up and connection with clinical providers at specific ages demonstrate that most children are being seen by appropriate clinical providers at the age of three, using the 2019 birth cohort, for example, with 94% success.

## 4. Discussion

Since 2006, there has been a call for the NBS field to gather and analyze LTFU data to evaluate the impact of newborn screening. While many have worked towards this goal [2,4,6,7,8,9], it has become clear that there is a need to codify the minimal data elements to make data collection more feasible. Our team evaluated the literature and quickly concluded that it was too difficult to gather data to address all the critical questions posed [7,9,15]. Our team was able to address three questions that we propose as a foundation for LTFU—(1) is the child alive? (2) is the child receiving care for their condition? and (3) is the child receiving appropriate care for their condition? This basic insight also allows all stakeholder groups—public health programs, research groups, and clinical care providers—to look more closely at the data and determine if there is inequity in groups who are deceased or are not connected to care after diagnosis of a newborn screening condition. Furthermore, once a system is in place and the minimal LTFU data are being gathered, there is an opportunity to start exploring the other questions that may be more difficult to answer.

All six programs—four programs associated with state-wide newborn screening systems, a university research program, and a national research program—were able to provide LTFU data. The programs also differed in the structure of their LTFU systems, some requiring consent and tracking participating children similar to a registry. In contrast, others tracked all identified infants in the public health system, similar to a surveillance model. Overall, the results from our analysis provide a promising snapshot of the impact of NBS. In 2022, 83.8% of the children in these LTFU programs were alive and still living in the jurisdiction of the follow-up system at the beginning of 2022; of those, 92.0% had contact with a specialist in 2022, and 87.7% were receiving appropriate care in 2022.

Determining the well-being of children identified with newborn screening has been a topic of discussion since the founding advisory committee members started discussing a national recommendation panel [16,17]. The LTFU data presented here provide evidence that children can be identified through public health long-term follow-up systems, and the vast majority are being followed by the appropriate clinical providers and meeting the recommended guidelines for follow-up. The approach presented here could provide the structure for the minimal elements for all LTFU programs. It could be applied to core RUSP disorders, secondary disorders, or other disorders being screened for or piloted in states. Moving forward, the challenge for the public health system is to identify those not receiving the appropriate care and seek solutions to any barriers that families may encounter. The next step is to expand the LTFU data collection to more newborn screening programs; this will help determine the feasibility of the minimal LTFU data elements proposed and could inform discussions around setting benchmarks for the rate/goals for each outcome. Once these LTFU data elements have been tested and refined so a majority of LTFU programs in the US can provide data, it would be helpful to identify other data elements that intersect with public health, research, and clinical care (see Figure 1) that can be gathered consistently, reliably, and with minimal effort. When looking at Figure 1, a potential place to start the conversation might be around healthcare utilization, including access to a medical home.

### Limitations

There are a few limitations to this project. First, two programs are consent-based, so their data may not be representative of their populations. For the UCSF program, follow-up rates are high. Patients who have not consented may receive care outside of the seven Immunology Centers of Excellence in the state and so are not reflected in our report. Furthermore, at UCSF, they currently do not have connections to vital records to be able to record if a child has died, so their ability to accurately assess the proportion of children who died in the previous year is limited.

For other programs, the data rely on children within a healthcare system, potentially excluding those not connected to any healthcare system, leading to the potential underrepresentation of specific demographic groups. Most programs did not have a process for tracking patients who had moved their care out of state or to a provider not affiliated with the state NBS. Future efforts may need to establish a national system to follow patients who relocate and move their care out of the jurisdiction.

Another notable limitation in determining if patients have had a visit in the past 12 months is the restricted connectivity of PCPs and specialists to a health information exchange (HIE). It can be particularly difficult to collect these data from PCP sites that still rely on paper records or non-interfacing electronic health record (EHR) systems with HIEs. However, this limitation could decrease over time as more practices adopt electronic health records, and HIEs become more interoperable. Lastly, the sample size is small, as data collection focused on specific conditions over a limited timeframe. There is potential for a more comprehensive understanding of the impact of newborn screening with an expanded data collection effort over time.

## 5. Conclusions

While these minimal LTFU data elements are insufficient to fully answer the question about the impact of NBS on public health, they are a feasible starting point. Ensuring that children are followed by care providers and are receiving appropriate care as established within their states is reassuring for public health professionals. It is not the responsibility of public health to monitor changes in clinical care or to ensure a child is following all the clinical recommendations for care. Instead, it is the duty of the public health system to confirm that children with a disorder diagnosed as the result of an abnormal NBS have access to appropriate care [11], and when children are not receiving care to identify and remove barriers. This should be seen as the responsibility of the public health *system* and not just the newborn screening programs, as ensuring access to care can be a monumental task requiring all interested parties’ input.

## Figures and Tables

**Figure 1 IJNS-10-00037-f001:**
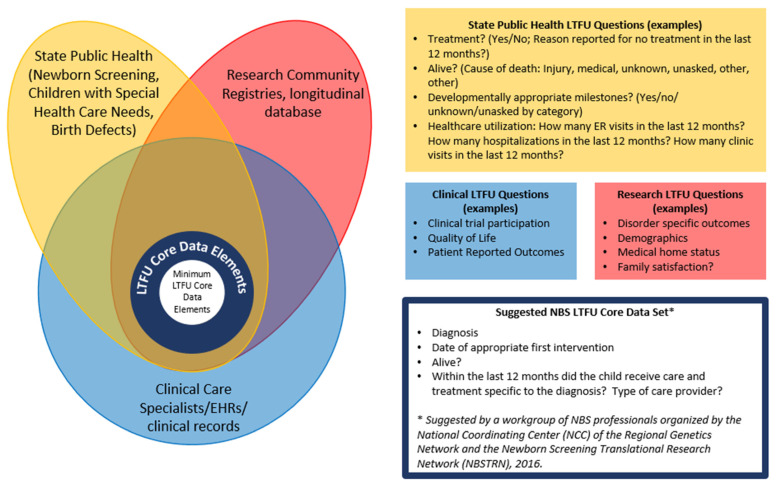
LTFU stakeholders and potential LTFU.

**Table 1 IJNS-10-00037-t001:** Focus, data sources and data limitations for each of the six LTFU awardees, representing different approaches, partnerships, and disorders studied.

Program	LTFU Program Focus (Goal, Conditions, Time Period)	Data Source(s)	Data Limitations
CO/WY	To ensure all children identified through newborn screening in Colorado/Wyoming receive appropriate follow-up for their disorders and to identify barriers leading to a child not receiving appropriate care. The program tracks all dried blood spot newborn screened conditions, 2002—present, except for congenital hypothyroidism (CH); these children will be added in future iterations.	NBS Dashboard in EPIC Clinic data EHR (EPIC)	Follow-up with specialists outside of the primary children’s hospital may be missed.
CT	To ensure timely and appropriate follow-up care for people diagnosed with a condition through newborn screening in CT. The network emphasizes comprehensive care coordination for optimal long-term outcomes by utilizing electronic health record-based registries and dashboards. The Network’s LTFU registry currently tracks patients identified with a condition through newborn screening in CT since March 1, 2019, except for cystic fibrosis (CF), critical congenital heart disease (CCHD), or hearing screen; separate programs follow those patients.	Clinic data EHR (EPIC) Epic Care Everywhere Health Information Exchange	Follow-up with specialists outside of the primary children’s hospital can be missed if not sent to health information exchange
NY	To develop a sustainable infrastructure to expand the newborn screening LTFU patient registry to include all the inherited metabolic disorders (IMD) on the newborn screening panel.	Lab Information System Electronic Medical Record (EHR) Data System from Specialty Centers	Need Informed Consent
ND	To ensure that newborns and children identified through newborn screening (NBS) achieve the best possible outcomes by utilizing a comprehensive model of LTFU that demonstrates collaborations between clinicians, public health agencies and families to create a system of care that can assess and coordinate follow-up and treatment of newborn screening conditions.	Vital Records Care Coordination Module within the North Dakota Health Information Network (NDHIN) NBS LTFU records	Starting screening for Pompe/MPSI in 2024, limited data for SMA
UCSF	To design and implement a comprehensive, family-centered LTFU program that becomes the standard for following clinical outcomes, supporting child and caregiver well-being, and anticipating future needs of children with Severe Combined Immunodeficiency (SCID) and T-cell lymphopenia (TCL) disorders.	A list of all patients with a positive NBS for SCID and <150 t-cell count from the California Department of Public Health EHR (EPIC)	Need Informed Consent and unable to report on deceased patients.
ACMG	To develop a comprehensive LTFU model system that demonstrates collaborations between clinicians, public health agencies, and families and assures the best possible outcomes for individuals identified through newborn screening. The project uses spinal muscular atrophy (SMA) as a model collecting data on cases within the first five years of life, engaging with up to five clinical sites, and reporting annual, de-identified aggregated data to state programs through the use of online dashboards. The type and scope of data collected were informed by parents and families with a family member who has SMA.	Pediatric neurologists reported newborns and children diagnosed with SMA following NBS.	Only retrospective data based on a REDCap survey with 81 questions (53/81 longitudinal) with up to five years of life per case; minimum of one case per center

**Table 2 IJNS-10-00037-t002:** Five final minimal long-term final data elements.

Variable	Definition
Number Of Children Diagnosed as Having the Condition	The condition was screened using NBS dried blood spot. The NBS was either a true positive or a false negative. The child was confirmed to have the condition following the NBS.
Number Of Children Who Died or Moved Before the Year Reported	The child died before January 1st for the year being reported. The child moved out of the jurisdiction of the program (e.g., NBS, clinic) providing the LTFU data before January 1st of the year being reported.
Number Of Children with Condition Who Are Alive	Child is not classified as deceased in the EHR; or Child is not classified as deceased in the state vistal record system; or Child is not reported as deceased to the LTFU program.
Number Of Children Who Had Contact with A Specialist for Their Disorder Within the Last 12 Months	There is a record that the child/family met with the specialist in person, via telehealth, or on a phone call; or There is a record that the child/family communicated with the specialist using email.
Number Of Children Who Received Appropriate Care Specific to The Diagnosis Within the Last 12 Month	The number of children who saw the appropriate specialist on the recommended cadence for care within the state/jurisdiction. The definitions of appropriate care were derived using a crosswalk between the recommended visit cadence from the participating programs, which were informed by practitioners and care guidelines at the jurisdictional level.

**Table 3 IJNS-10-00037-t003:** The proposed data elements can be used to assess long-term follow-up across multiple programs, groups of disorders, and birth cohorts; data from six LTFU awardees are presented.

Birth Cohort	Diagnosis as Determined by Published Case Definitions [15]	Children Known to Be Alive and Living in the Jurisdiction at the Beginning of 2022/Number of Children with Disorder	Children Who Had at Least One Contact with Specialist in 2022/Number of Children Known to Be Alive and in Jurisdiction at the Beginning of 2022	Children Receiving Appropriate Care in 2022/Number of Children Known to Be Alive and in Jurisdiction at the Beginning of 2022
All Birth Cohorts	All NBS Disorders Reported	563/672 (83.8%)	518/563 (92.0%)	494/563 (87.7%)
2018	All NBS Disorders Reported	67/91 (73.6%)	57/67 (85.1%)	55/67 (82.1%)
2019		100/132 (75.8%)	94/100 (94.0%)	94/100 (94.0%)
2020		129/155 (83.2%)	121/129 (93.8%)	114/129 (88.4%)
2021		139/155 (89.7%)	124/139 (89.2%)	116/139 (83.5%)
2022		128/139 (92.1%)	122/128 (95.3%)	115/128 (89.8%)
2018	Metabolic Conditions	20/24 (83.3%)	15/20 (75.0%)	15/20 (75.0%)
2019		18/24 (75.0%)	14/18 (77.8%)	14/18 (77.8%)
2020		29/34 (85.3%)	26/29 (89.7%)	20/29 (69.0%)
2021		38/40 (95.0%)	29/38 (76.3%)	25/38 (65.8%)
2022		28/29 (96.6%)	24/28 (85.7%)	21/28 (75.0%)
2018	Congenital Adrenal Hyperplasia	8/10 (80.0%)	8/8 (100.0%)	7/8 (87.5%)
2019		8/11 (72.7%)	8/8 (100.0%)	8/8 (100.0%)
2020		8/8 (100.0%)	7/8 (87.5%)	7/8 (87.5%)
2021		10/10 (100.0%)	9/10 (90.0%)	8/10 (80.0%)
2022		6/6 (100.0%)	6/6 (100.0%)	6/6 (100.0%)
2018	Congenital Hypothyroidism	None reported	None reported	None reported
2019		16/26 (61.5%)	16/16 (100.0%)	16/16 (100.0%)
2020		29/37 (78.4%)	29/29 (100.0%)	29/29 (100.0%)
2021		32/39 (82.1%)	32/32 (100.0%)	32/32 (100.0%)
2022		29/35 (82.9%)	29/29 (100.0%)	27/29 (93.1%)
2018	Hemoglobinopathies	10/19 (52.6%)	9/10 (90.0%)	8/10 (80.0%)
2019		19/26 (73.1%)	18/19 (94.7%)	18/19 (94.7%)
2020		22/29 (75.9%)	19/22 (86.4%)	18/22 (81.8%)
2021		18/21 (85.7%)	15/18 (83.3%)	13/18 (72.2%)
2022		20/21 (95.2%)	20/20 (100.0%)	18/20 (90.0%)
2018	Cystic Fibrosis	17/21 (81.0%)	16/17 (94.1%)	16/17 (94.1%)
2019		23/27 (85.2%)	23/23 (100.0%)	23/23 (100.0%)
2020		21/22 (95.5%)	21/21 (100.0%)	21/21 (100.0%)
2021		14/17 (82.4%)	13/14 (92.9%)	13/14 (92.9%)
2022		20/21 (95.2%)	20/20 (100.0%)	20/20 (100.0%)
2018	Severe Combined Immunodeficiency (SCID)	6/11 (54.5%)	3/6 (50.0%)	3/6 (50.0%)
2019		6/8 (75.0%)	5/6 (83.3%)	5/6 (83.3%)
2020		11/14 (78.6%)	10/11 (90.9%)	10/11 (90.9%)
2021		4/4 (100.0%)	3/4 (75.0%)	3/4 (75.0%)
2022		9/10 (90.0%)	7/9 (77.8%)	7/9 (77.8%)
2018	Non-SCID T-cell lymphopenia	4/4 (100.0%)	4/4 (100.0%)	4/4 (100.0%)
2019		4/4 (100.0%)	4/4 (100.0%)	4/4 (100.0%)
2020		1/1 (100.0%)	1/1 (100.0%)	1/1 (100.0%)
2021		4/4 (100.0%)	4/4 (100.0%)	4/4 (100.0%)
2022		8/8 (100.0%)	8/8 (100.0%)	8/8 (100.0%)
2018	Spinal Muscular Atrophy	2/2 (100.0%)	2/2 (100.0%)	2/2 (100.0%)
2019		6/6 (100.0%)	6/6 (100.0%)	6/6 (100.0%)
2020		8/10 (80.0%)	8/8 (100.0%)	8/8 (100.0%)
2021		19/20 (95.0%)	19/19 (100.0%)	18/19 (94.7%)
2022		8/9 (88.9%)	8/8 (100.0%)	8/8 (100.0%)

## Data Availability

Data are available upon request to the primary (yvonne.kellar-guenther@cphinnovation.org) or senior author (marci.sontag@cphinnovation.org).

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
