# Peer review of "Defining the Minimal Long-Term Follow-Up Data Elements for Newborn Screening"

_2409-515X, 2024, doi:10.3390/ijns10020037_

Round 1

Reviewer 1 Report

Comments and Suggestions for Authors

General comments

This paper makes a valuable contribution by presenting a set of three minimal data elements for public health newborn screening (NBS) long-term follow-up (LTFU) programs with clinical involvement. It further demonstrates the feasibility of implementation of this approach by four US public health NBS programs and reports outcome information pooled from those four programs and two clinical LTFU programs. 

The paper could benefit from additional clarification and explanation of terms and concepts and improved citation of references. In particular, the meaning and intent of NBS disorders and NBS LTFU should be clearly defined and elucidated to minimize potential confusion. NBS can refer to both clinical services and a public health program. The focus of the paper is on public health NBS and, more specifically, NBS that is conducted through laboratory analyses of dried blood spot (DBS) specimens. In the United States public health DBS NBS is mandated on the rationale that early treatment is required to prevent irreversible damage or death, at least for a subset of core NBS disorders. The conventional rationale for NBS LTFU is to assure that infants and families identified through mandated screening receive recommended treatments, as the authors acknowledge at L84-85. A different kind of LTFU program involves establishing a patient registry for NBS disorders that includes children with diagnoses, regardless of whether they were identified through NBS. This last approach is implicit in the proposed framework, which includes cases not detected through NBS. An explicit rationale for establishing patient registries for NBS disorders should be provided that is distinct from the conventional rationale of NBS LTFU.

The authors should also clearly distinguish between core or primary NBS disorders for which screening is mandated based on evidence of benefit from early identification and secondary NBS disorders that are detected incidentally through screening for primary NBS disorders. The introduction notes, “there are 37 core conditions and 26 secondary conditions on the Recommended Uniform Screening Panel (RUSP)” but does not acknowledge the differences between the two lists. That is crucial, because for many people, including HRSA, the RUSP is understood to refer to the list of core conditions, which are the only conditions that are federally recommended for screening. According to Recommended Uniform Screening Panel | HRSA, core RUSP conditions are those “Disorders that should be included in every Newborn Screening Program.” Secondary conditions, in contrast, are “Disorders that can be detected in the differential diagnosis of a core disorder.” HRSA further states, “The RUSP is a list of disorders that the Secretary of the Department of Health and Human Services (HHS) recommends for states to screen as part of their state universal newborn screening (NBS) programs. Disorders on the RUSP are chosen based on evidence that supports the potential net benefit of screening, the ability of states to screen for the disorder, and the availability of effective treatments. It is recommended that every newborn be screened for all disorders on the RUSP. Most states screen for the majority of disorders on the RUSP; newer conditions are still in process of adoption. Some states also screen for additional disorders. Although states ultimately determine what disorders their NBS program will screen for, the RUSP establishes a standardized list of disorders that have been supported by the Advisory Committee on Heritable Disorders in Newborns and Children and recommended by the Secretary of HHS.” Those statements apply only to the core conditions on the RUSP and do not apply to the list of secondary conditions, none of which were chosen based on evidence of net benefit nor recommended to states to be screened.

Specific comments

L22-23: The text vaguely states, “little is known about what happens to these children after diagnosis”, which is ambiguous since “these children” are not defined. The first part of the sentence refers to NBS, not “children”.  LTFU can refer to the ongoing monitoring of children who are diagnosed following a positive NBS result. In contrast, the authors include in NBS LTFU children who were diagnosed clinically unrelated to NBS, i.e., did not screen positive (false-negative results or not screened). A rationale for that inclusion should be presented.

L33-34: “In 2022, about 83.8% (563/672) of the children in these LTFU programs were alive, 92.0% saw a specialist, and 87.7% received appropriate care.” The meaning of the denominator of 672 infants is ambiguous. Were children who died prior to January 1, 2022 included, as is implied by the text at L255-257, or were they excluded, as is implied by the text at L207-210? Also, were the 563 children who were alive and served as the denominator for the other two measures alive at the beginning of 2022 or at the end of 2022? Also, it appears from the text that 92.0% of the 563 children had ever seen a specialist, not necessarily during 2022; that should be made explicit here.

L84-90: This paragraph relies heavily on reference 13, an unpublished report that is cited four times in four sentences. Unfortunately, that report is not accessible; the URL is a broken link and no report with that name can be found by searching the NewSTEPS website. It is essential that the authors “

L85-86: The assertion that the duration of LTFU can be “the entire lifespan” is unsupported by references 14 and 15, both of which suggest that an appropriate endpoint for public health LTFU surveillance is the transition from pediatric care to adult care.

L88-90: This appears to be a false dichotomy. Public health is likewise interested in overall health and developmental outcomes.

L93-99: The authors should recheck the cited references for appropriateness. For example, it does not appear that reference 1 listed being alive as an outcome.

L133-137: The selection of the ACMG program for this project does not clearly correspond to the stated purpose to provide services to “families of newborns and children diagnosed with a condition as a result of an abnormal newborn screen.” Unlike the other five collaborating programs, the ACMG does not begin with a cohort of children diagnosed with a condition as a result of NBS. Instead, that project appears to be a clinical registry of pediatric patients diagnosed with SMA and seen by pediatric neurologists.

L147-149: Reference 16 lists a nonworking URL. Since that resource is unavailable, it would be helpful if the authors were to include a summary of the content of the missing report.  

L196-202: It is unclear whether the UCSF project was able to provide information on one of the three minimal data elements, namely whether children had died. Table 2 states that the UCSF project was “unable to report on deceased patients.”

L207: It is interesting that cases not detected by NBS programs are included in the NBS LTFU denominator. How many such cases were identified? What are the implications for combined those cases with children identified through NBS?

L207-210: The exclusion of children who died or moved away prior to 2022 requires clarification. If those children were excluded from the denominator for the calculation of the survival outcome it would mean that the outcome measure refers to deaths during the most recent calendar year rather than cumulative deaths since enrollment in LTFU. If the authors meant that the children who were still alive and present on January 1, 2022 were used as the denominator for the other two outcome measures that should be so stated.

L214-217: This list comprises both primary and secondary RUSP conditions, the latter of which are not clearly defined.

Notably, “hemoglobinopathies” presumably includes sickle cell disease, which is listed on the RUSP as three distinct primary conditions, and “various other hemoglobinopathies” an ill-defined category that is included in the RUSP secondary conditions list. The RUSP website singles out hemoglobin C, D, and E disorders as well as “genetic trait”, i.e., a carrier state, which is not a disorder. No mention is made of alpha or beta thalassemias, which are also hemoglobinopathies. R reference 18 refers readers to the online NewSTEPS list of case definitions, https://www.newsteps.org/nbs-disorders/case-definitions?q=case-definitions. NewSTEPS has no “hemoglobinopathies” category but under “Sickle cell diseases” lists hemoglobin C, D, and E disorders, disorders that are instead listed as “various other hemoglobinopathies” on the RUSP list of secondary conditions.  

T-cell lymphopenia does not comprise a single disorder. The secondary conditions list for the RUSP includes “T-cell related lymphocyte deficiencies”, which are referred to as “a group of inherited (genetic) conditions where a baby is born with a decreased immune system.” Since SCID, which is a primary RUSP disorder, entails T-cell deficiency, this category is also referred to as “Non-SCID primary immunodeficiencies or other conditions associated with low T-cells” to avoid double counting with SCID. Reference 18 and the NewSTEPs site list four subtypes of SCID: typical SCID, leaky SCID, Omenn syndrome, and “Non-SCID conditions associated with SCID NBS”, which include Syndromes with low T-cell number, Secondary T-cell lymphopenia, preterm birth alone, and idiopathic T-cell lymphopenia (formerly called variant SCID). It would be helpful if the authors were to list which specific disorders were included under hemoglobinopathies and T-cell lymphopenia.

L278-280: This sentence incorrectly cites reference 6 in place of 16, not that it makes any practical difference since reference 16 is unavailable.

L371: The institutional author for reference 6 should be displayed the same as the PubMed citations, PMID: 21697806

Author Response

Dear Reviewer One:

First off, thank you so much for taking the time and providing such thoughtful comments and suggestions. It is always helpful to see where our ideas are lacking clarity. Your feedback has pushed us to be more specific and has made this paper stronger.

One key area that this review helped us to solidify was where the data was extracted from and who the responsible parties for LTFU are. The collaboration across the six programs highlights that the programs can be managed by different types of entities, yet these data elements can successfully be used by all of the entities to assess LTFU.  We have tried to reframe our write up to reflect that.

In addition, we knew when we wrote this that there was never an adoption by the ACHDNC or even the ACHDNC FUTR workgroup of the core data elements. As you saw from our previous version, the only evidence we had was a slide (slide #4) from the workgroup report out at the April 24, 2019, ACHDNC meeting. Two of our authors were part of this work group. As we worked on this revision, there was a realization that the questions came from a workgroup organized by the National Coordinating Center (NCC) and NBSTRN. Again, there was no formal write up, but we do cite personal communication with a member of NBSTRN who was instrumental in the meeting. We now state in the paper that the NBS LTFU Core Data Set our group started with was suggested in 2016 by a workgroup of NBS professionals organized by the National Coordinating Center (NCC) of the Regional Genetics Network and the Newborn Screening Translational Research Network (NBSTRN).

Another major change, as suggested by reviewer 3, was to shorten the literature review and tighten up sections 3.1 and 3.1.2. We have made those changes and hope that we have still provided enough context for the readers.

Finally, we realized during the re-write that we needed to be specific that we were speaking about US newborn screening and have tried to clarify that as well.

We address each specific recommendation below. 

Thank you again for taking the time to review this article.

General comments

This paper makes a valuable contribution by presenting a set of three minimal data elements for public health newborn screening (NBS) long-term follow-up (LTFU) programs with clinical involvement. It further demonstrates the feasibility of implementation of this approach by four US public health NBS programs and reports outcome information pooled from those four programs and two clinical LTFU programs. 

  1. The paper could benefit from additional clarification and explanation of terms and concepts and improved citation of references. In particular, the meaning and intent of NBS disorders and NBS LTFU should be clearly defined and elucidated to minimize potential confusion. NBS can refer to both clinical services and a public health program. The focus of the paper is on public health NBS and, more specifically, NBS that is conducted through laboratory analyses of dried blood spot (DBS) specimens.

Thank you for highlighting the distinctions between clinical services for NBS and a public health program. The six groups funded through this program were charged with creating models of LTFU and as noted on lines 137 to 140, “The purpose of the program is to support comprehensive models of long-term follow-up that demonstrate collaborations between clinicians, public health agencies, and families” (HRSA-21-079 funding opportunity announcement, pg. i). Each of the six programs approached this in different ways but the program was not intended to only focus on DBS NBS in public health. Table 1 presents brief descriptions of each program.

We also edited the description of the denominator to highlight that we focused on disorders that are part of the NBS DBS screening and that false negatives are included in our numbers. The section reads: 3.1.1 Denominator. For the minimal LTFU data elements, the denominator represents the number of children within the birth cohort who have been diagnosed as having a condition that was screened for using NBS dried blood spots, including those cases who were diagnosed after a false negative NBS, minus those who died or moved their care out of the jurisdiction before the year being reported (i.e., in our case, those who died or moved before January 1, 2022).

  1. In the United States public health DBS NBS is mandated on the rationale that early treatment is required to prevent irreversible damage or death, at least for a subset of core NBS disorders. The conventional rationale for NBS LTFU is to assure that infants and families identified through mandated screening receive recommended treatments, as the authors acknowledge at L84-85. A different kind of LTFU program involves establishing a patient registry for NBS disorders that includes children with diagnoses, regardless of whether they were identified through NBS. This last approach is implicit in the proposed framework, which includes cases not detected through NBS. An explicit rationale for establishing patient registries for NBS disorders should be provided that is distinct from the conventional rationale of NBS LTFU.

The reviewer brings up an excellent point. Registries are a method for tracking affected individuals over time and typically require consent from participants.  Some of the awardees followed a model that is consistent with that, while others proposed an approach that is more similar to public health surveillance models. However, the data elements presented in this manuscript can be used regardless of the structure of the LTFU system.  

We added the following sentence to the second paragraph of the discussion:

“The programs also differed in the structure of their LTFU systems, some requiring consent and tracking participating children similar to a registry, while others tracked all identified infants in the public health system similar to a surveillance model. “

  1. The authors should also clearly distinguish between core or primary NBS disorders for which screening is mandated based on evidence of benefit from early identification and secondary NBS disorders that are detected incidentally through screening for primary NBS disorders. The introduction notes, “there are 37 core conditions and 26 secondary conditions on the Recommended Uniform Screening Panel (RUSP)” but does not acknowledge the differences between the two lists. That is crucial, because for many people, including HRSA, the RUSP is understood to refer to the list of core conditions, which are the only conditions that are federally recommended for screening. According to Recommended Uniform Screening Panel | HRSA, core RUSP conditions are those “Disorders that should be included in every Newborn Screening Program.” Secondary conditions, in contrast, are “Disorders that can be detected in the differential diagnosis of a core disorder.” HRSA further states, “The RUSP is a list of disorders that the Secretary of the Department of Health and Human Services (HHS) recommends for states to screen as part of their state universal newborn screening (NBS) programs. Disorders on the RUSP are chosen based on evidence that supports the potential net benefit of screening, the ability of states to screen for the disorder, and the availability of effective treatments. It is recommended that every newborn be screened for all disorders on the RUSP. Most states screen for the majority of disorders on the RUSP; newer conditions are still in process of adoption. Some states also screen for additional disorders. Although states ultimately determine what disorders their NBS program will screen for, the RUSP establishes a standardized list of disorders that have been supported by the Advisory Committee on Heritable Disorders in Newborns and Children and recommended by the Secretary of HHS.” Those statements apply only to the core conditions on the RUSP and do not apply to the list of secondary conditions, none of which were chosen based on evidence of net benefit nor recommended to states to be screened.

Thank you for this comment.  We chose disorders for this report based on the disorders that each LTFU awardee had developed within their programs. We did not prescribe what needed to be collected. The data elements presented here could be used as the minimal elements suggested for all LTFU programs, and could be used for Core RUSP disorders, secondary disorders, or disorders that are being piloted in states.  

Please note we took out the line about the RUSP conditions in the introduction.

We have added the following sentences to the discussion to clarify this point (the last paragraph prior to the limitations):

“The approach presented here could provide the structure for the minimal elements for all LTFU programs. It could be applied to Core RUSP disorders, secondary disorders, or other disorders that are being screened for or piloted in states.”

Specific comments

  1. L22-23: The text vaguely states, “little is known about what happens to these children after diagnosis”, which is ambiguous since “these children” are not defined. The first part of the sentence refers to NBS, not “children”. LTFU can refer to the ongoing monitoring of children who are diagnosed following a positive NBS result. In contrast, the authors include in NBS LTFU children who were diagnosed clinically unrelated to NBS, i.e., did not screen positive (false-negative results or not screened). A rationale for that inclusion should be presented.

Reworded line 22-23 to read “Newborn screening (NBS) is hailed as a public health success, but little is known about what happens to the children diagnosed following a positive NBS.”

Children who have false-negative newborn screening results within a state are also a part of the broader public health system and should receive the same attention to care and long-term follow-up as those with true-positive NBS DBS results. We do not believe that there are any additional implications for combining these cases. We did not differentiate cases that were true positives vs. false negative in our data collection or our presentation as the performance of the various NBS systems and algorithms is beyond the scope of our paper.  

We appreciate the reviewer’s attention to detail.  We have edited the abstract to be clearer. now reading:

“In 2022, about 83.8% (563/672) of the children in these LTFU programs were alive and still living in the jurisdiction, 92.0% (518/563) of those saw a specialist, and 87.7% (494/563) received appropriate care. “

Further, we edited lines 255-257 to now read:

“Some children moved their care out of the jurisdiction or died prior to 2022, leaving 563 eligible for LTFU at the beginning of the reporting year (2022).”

We have also added clarification to the Table to indicate that eligible children were alive and living in the jurisdiction in 2022, and that specialist visits were also specific to 2022.  Finally, we edited the discussion to reflect this as follows:

“In 2022, 83.8% of the children in these LTFU programs were alive and still living in the jurisdiction of the follow-up system at the beginning of 2022; of those, 92% of those had contact with a specialist in 2022, and 87.7% were receiving appropriate care in 2022. “

  1. L84-90: This paragraph relies heavily on reference 13, an unpublished report that is cited four times in four sentences. Unfortunately, that report is not accessible; the URL is a broken link and no report with that name can be found by searching the NewSTEPS website. It is essential that the authors “

We took out a lot of this paragraph. We did leave in the first two lines though and only included published articles. We took out the unpublished report from APHL. Sorry that link did not work for you.

  1. L85-86: The assertion that the duration of LTFU can be “the entire lifespan” is unsupported by references 14 and 15, both of which suggest that an appropriate endpoint for public health LTFU surveillance is the transition from pediatric care to adult care.

We respectfully disagree with you on this. We have left in the line “The length of NBS LTFU has been proposed to vary from diagnosis up until school age, to 18 years of life, or throughout the entire lifespan [5, 6].” Below are direct quotes from the articles cited to back up our claim of LTFU potentially lasting a lifetime. We understand ACHDNC has determined their jurisdiction ends when a person becomes a legal adult but that is not what we were arguing.

“Newborn screening (NBS) long-term follow-up (LTFU) begins on confirmed diagnosis of a disorder1 and may continue throughout life.” (p. 484) and “Ideally, the need for LTFU spans from birth to adulthood,

encompassing preconception and prenatal care for women. The length of time designated for LTFU is often determined by the rules and regulations of the state and by available resources.” (p. 489), Hinton, C. F., C. T. Mai, S. K. Nabukera, L. D. Botto, L. Feuchtbaum, P. A. Romitti, Y. Wang, K. N. Piper, and R. S. Olney. "Developing a Public Health-Tracking System for Follow-up of Newborn Screening Metabolic Conditions: A Four-State Pilot Project Structure and Initial Findings." Genet Med 16, no. 6 (2014): 484-90.)

“The time frame for long-term follow-up is the lifespan of the affected individual; however, the responsibility of the ACHDGDNC as set by its authorizing legislation is from birth to age 21 years.” (p. 260, Kemper, A. R., C. A. Boyle, J. Aceves, D. Dougherty, J. Figge, J. L. Fisch, A. R. Hinman, C. L. Greene, C. A. Kus, J. Miller, D. Robertson, J. Telfair, B. Therrell, M. Lloyd-Puryear, P. C. van Dyck, and R. R. Howell. "Long-Term Follow-up after Diagnosis Resulting from Newborn Screening: Statement of the Us Secretary of Health and Human Services' Advisory Committee on Heritable Disorders and Genetic Diseases in Newborns and Children." Genet Med 10, no. 4 (2008): 259-61.

  1. L88-90: This appears to be a false dichotomy. Public health is likewise interested in overall health and developmental outcomes.

We removed this line from the write up.

  1. L93-99: The authors should recheck the cited references for appropriateness. For example, it does not appear that reference 1 listed being alive as an outcome.

We agree with this statement. Bailey is laying out what are meaningful outcomes for NBS, and listed number of lives saved (p. 11), but he doesn’t not say this is a metric that should be measured for LTFU. Thank you for catching this error. We also removed reference to this article in this section since the author is not making recommendations around LTFU measures.

We have checked our other references and feel comfortable leaving them in.

  1. L133-137: The selection of the ACMG program for this project does not clearly correspond to the stated purpose to provide services to “families of newborns and children diagnosed with a condition as a result of an abnormal newborn screen.” Unlike the other five collaborating programs, the ACMG does not begin with a cohort of children diagnosed with a condition as a result of NBS. Instead, that project appears to be a clinical registry of pediatric patients diagnosed with SMA and seen by pediatric neurologists.

The reviewers bring up an interesting point. We believe that there isn’t a one size fits all approach to LTFU. The ACMG program only enrolls newborns and children who were screen positive for SMA and diagnosed with SMA because of a positive newborn screen. Please review the ACMG program description in Table 1 which states: “….for individuals identified through newborn screening.”

Finally, while the ACMG approach may differ from the other programs, the data elements proposed worked for that LTFU system as well, supporting the underlying tenet of our work.   

  1. L147-149: Reference 16 lists a nonworking URL. Since that resource is unavailable, it would be helpful if the authors were to include a summary of the content of the missing report.

We removed this reference. As we worked on this revision, there was a realization that the questions came from a workgroup organized by the National Coordinating Center (NCC) and NBSTRN. There was no formal write up, but we do cite personal communication with a member of NBSTRN who was instrumental in the meeting. We now state in the paper that the NBS LTFU Core Data Set our group started with was suggested by in 2016 by a workgroup of NBS professionals organized by the National Coordinating Center (NCC) of the Regional Genetics Network and the Newborn Screening Translational Research Network [NBSTRN]. We are unable to provide a link to any reference however because none seems to exist.

  1. L196-202: It is unclear whether the UCSF project was able to provide information on one of the three minimal data elements, namely whether children had died. Table 2 states that the UCSF project was “unable to report on deceased patients.”

The reviewer is correct.  UCSF was unable to provide information on deceased patients for the element “Number of children who moved or died prior to the follow-up year at this time.  They were able to report the number of children from each birth cohort who were known to be alive and living in their jurisdiction, and the remainder of the indicators of individuals receiving follow-up.  We have added the following to the limitations to reflect this.

“Further, at UCSF, they currently do not have connections to vital records to be able to record if a child has died, so their ability to accurately assess the proportion of children who died in the previous year is limited.”

  1. L207: It is interesting that cases not detected by NBS programs are included in the NBS LTFU denominator. How many such cases were identified? What are the implications for combined those cases with children identified through NBS?

As we stated earlier, children who have false-negative newborn screening results within a state are also a part of the broader public health system and should receive the same attention to care and long-term follow-up as those with true-positive NBS DBS results. We do not believe that there are any additional implications for combining these cases. We did not differentiate cases that were true positives vs. false negative in our data collection or our presentation as the performance of the various NBS systems and algorithms is beyond the scope of our paper.  

  1. L207-210: The exclusion of children who died or moved away prior to 2022 requires clarification. If those children were excluded from the denominator for the calculation of the survival outcome it would mean that the outcome measure refers to deaths during the most recent calendar year rather than cumulative deaths since enrollment in LTFU. If the authors meant that the children who were still alive and present on January 1, 2022 were used as the denominator for the other two outcome measures that should be so stated.

We excluded the children who moved or died prior to 2022 as the intent of our data collection was to determine how well the system works to track children known to the system in any given year. Once a child is deceased or has moved the public health system of the child’s birth state is not responsible for the long-term follow-up of that child. The metrics that are calculated for 2022 are to assess if children who are alive and known to the state are receiving care for their disorders.  Hopefully this is clearer in this version.

  1. L214-217: This list comprises both primary and secondary RUSP conditions, the latter of which are not clearly defined.

Notably, “hemoglobinopathies” presumably includes sickle cell disease, which is listed on the RUSP as three distinct primary conditions, and “various other hemoglobinopathies” an ill-defined category that is included in the RUSP secondary conditions list. The RUSP website singles out hemoglobin C, D, and E disorders as well as “genetic trait”, i.e., a carrier state, which is not a disorder. No mention is made of alpha or beta thalassemias, which are also hemoglobinopathies. R reference 18 refers readers to the online NewSTEPS list of case definitions, https://www.newsteps.org/nbs-disorders/case-definitions?q=case-definitions. NewSTEPS has no “hemoglobinopathies” category but under “Sickle cell diseases” lists hemoglobin C, D, and E disorders, disorders that are instead listed as “various other hemoglobinopathies” on the RUSP list of secondary conditions.

T-cell lymphopenia does not comprise a single disorder. The secondary conditions list for the RUSP includes “T-cell related lymphocyte deficiencies”, which are referred to as “a group of inherited (genetic) conditions where a baby is born with a decreased immune system.” Since SCID, which is a primary RUSP disorder, entails T-cell deficiency, this category is also referred to as “Non-SCID primary immunodeficiencies or other conditions associated with low T-cells” to avoid double counting with SCID. Reference 18 and the NewSTEPs site list four subtypes of SCID: typical SCID, leaky SCID, Omenn syndrome, and “Non-SCID conditions associated with SCID NBS”, which include Syndromes with low T-cell number, Secondary T-cell lymphopenia, preterm birth alone, and idiopathic T-cell lymphopenia (formerly called variant SCID). It would be helpful if the authors were to list which specific disorders were included under hemoglobinopathies and T-cell lymphopenia.

The reviewer is correct. The list does include Core and Secondary conditions. The intent of this manuscript was not to provide a comprehensive description of LTFU for all disorders, but rather to demonstrate that the metrics could be used for all disorders.  

We have added the following sentences to the discussion to clarify this point (the last paragraph prior to the limitations):

“The approach presented here could provide the structure for the minimal elements for all LTFU programs. It could be applied to Core RUSP disorders, secondary disorders, or other disorders that are being screened for or piloted in states.”

  1. L278-280: This sentence incorrectly cites reference 6 in place of 16, not that it makes any practical difference since reference 16 is unavailable.

Thank you for noting this. We have removed all references to ACHDNC FUTR slides (reference 16 in version 1).

  1. L371: The institutional author for reference 6 should be displayed the same as the PubMed citations, PMID: 21697806

Thank you. The citation now reads as: “Centers for Disease Control and Prevention (CDC). Ten great public health achievements--United States, 2001-2010. MMWR Morb Mortal Wkly Rep. 2011 May 20;60(19):619-23. PMID: 21597455.” This was the recommendation on the PubMed website.

Reviewer 2 Report

Comments and Suggestions for Authors

Summary

This manuscript describes the process whereby six newborn screening (NBS) “programs” agreed upon a “minimum needed data set” to collect long-term follow-up (LTFU) data from NBS programs across the US or the world to assess the effectiveness of NBS.  After reaching agreement on the data set, the organizations proceeded to collect those data from their programs and demonstrated the feasibility of doing so.

Strengths

As described by the authors, this manuscript represents the groundwork for starting to collect uniform LTFU data across the US. 

Areas for Improvement

Anyone reading this article will be familiar with the importance of LTFU.  The Introduction is about twice as long as it needs to be for this audience.

As two of the entities represented in this manuscript are not government run NBS programs, it is not clear whence they get their data.

It seems that “moved out of state” is lumped in as part of lost to follow-up.  Reporting that separately would be helpful because those who moved out of state are likely to be getting care elsewhere.  Lumping the two underestimates the effectiveness of LTFU.  The statement in the abstract that “83.8% (563/672) of the children in these 33 LTFU programs were alive” is alarming until reading the Results where the missing 16.2% either died OR moved.

I do not think Table 1 contributes to the value of the manuscript.

Minor Points

Ref 16 has a dead link.  I think the correct link is https://www.hrsa.gov/sites/default/files/hrsa/advisory-committees/heritable-disorders/meetings/day2-kus-follow-treatment.pdf  Accessed 3/15/24

This is a short slide set that does not seem to include the “data elements” referenced in the manuscript.  The slides only propose that minimal data elements be considered.  There may be a better reference from ACHDNC FUTR.

Sometime in the future it may be possible to pair NBS records with death certificates to get a better idea of mortality due to screened diseases.  The task is not trivial.

“Appropriate care” seems to be determined locally.  Some disorders have consensus guidelines from professional organizations and those should be used when available.

I do not understand why UCSF and ACMG consider themselves NBS programs.  I think the manuscript states that each of them has only a narrow focus of diseases represented in the data.

The yellow oval in Figure 1 should include NBS programs as part of “State Public Health”.

Author Response

Dear Reviewer Two:

First off, thank you so much for taking the time and providing such thoughtful comments and suggestions. It is always helpful to see where our ideas are lacking clarity. Your feedback has pushed us to be more specific and has made this paper stronger.

One key area that this review helped us to solidify was where the data was extracted from and who the responsible parties for LTFU are. The collaboration across the six programs highlights that the programs can be managed by different types of entities, yet these data elements can successfully be used by all of the entities to assess LTFU.  We have tried to reframe our write up to reflect that.

In addition, we knew when we wrote this that there was never an adoption by the ACHDNC or even the ACHDNC FUTR workgroup of the core data elements. As you saw from our previous version, the only evidence we had was a slide (slide #4) from the workgroup report out at the April 24, 2019, ACHDNC meeting. Two of our authors were part of this work group. As we worked on this revision, there was a realization that the questions came from a workgroup organized by the National Coordinating Center (NCC) and NBSTRN. Again, there was no formal write up, but we do cite personal communication with a member of NBSTRN who was instrumental in the meeting. We now state in the paper that the NBS LTFU Core Data Set our group started with was suggested in 2016 by a workgroup of NBS professionals organized by the National Coordinating Center (NCC) of the Regional Genetics Network and the Newborn Screening Translational Research Network (NBSTRN).

Another major change, as suggested by reviewer 3, was to shorten the literature review and tighten up sections 3.1 and 3.1.2. We have made those changes and hope that we have still provided enough context for the readers.

Finally, we realized during the re-write that we needed to be specific that we were speaking about US newborn screening and have tried to clarify that as well.

We address each specific recommendation below. 

Thank you again for taking the time to review this article.

Reviewer Two Comments:

summary

This manuscript describes the process whereby six newborn screening (NBS) “programs” agreed upon a “minimum needed data set” to collect long-term follow-up (LTFU) data from NBS programs across the US or the world to assess the effectiveness of NBS. After reaching agreement on the data set, the organizations proceeded to collect those data from their programs and demonstrated the feasibility of doing so.

Strengths

As described by the authors, this manuscript represents the groundwork for starting to collect uniform LTFU data across the US. 

Areas for Improvement

  1. Anyone reading this article will be familiar with the importance of LTFU. The Introduction is about twice as long as it needs to be for this audience.

Thank you for your feedback. We have shortened the introduction.

  1. As two of the entities represented in this manuscript are not government run NBS programs, it is not clear whence they get their data.

Four of the programs are directly associated with newborn screening programs, while only two of the entities represent NBS programs themselves (NY and ND). We believe that LTFU programs may not always need to be contained to the state public health department, by rather may be a collaboration between multiple entities. The variety of entities and partnerships presented here support that.  For all six of our entities, the data sources are listed in what is now Table 1 (Table 1 and 2 were merged as a part of this revision).

We also revised our description of the six awardees at the start of section 2 to read “Six awardees—four programs directly managed by or closely associated with state NBS systems (CT, CO/WY, NY, ND), one university program (UCSF), and one professional foundation (the American College of Medical Genetics (ACMG)) were funded by HRSA (HRSA-21-079) to “expand the ability of state public health agencies to provide screening, counseling and services” to the families of newborns and children diagnosed with a condition as a result of an abnormal newborn screen.”

  1. It seems that “moved out of state” is lumped in as part of lost to follow-up. Reporting that separately would be helpful because those who moved out of state are likely to be getting care elsewhere. Lumping the two underestimates the effectiveness of LTFU. The statement in the abstract that “83.8% (563/672) of the children in these 33 LTFU programs were alive” is alarming until reading the Results where the missing 16.2% either died OR moved.

We appreciate this reviewer’s comment. Our initial data element “Number of children who moved or died prior to the current follow-up year” is not intended to be lost to follow-up.  The count reflected in this measure reflects those who are not eligible for LTFU in the state at the beginning of the year. Therefore, we are subtracting them from our initial denominator in order to have an appropriate denominator for the other metrics.  Thank you for noting our error in the abstract as this is a very important note. We have corrected the abstract to read:

“In 2022, 83.8% (563/672) of the children in these LTFU programs were alive and living in the jurisdiction; of those, 92.0% (518/563) saw a specialist, and 87.7% (494/563) received appropriate care. “

  1. I do not think Table 1 contributes to the value of the manuscript.

We have combined Tables 1 and 2, and we believe the information provided supports different approaches to newborn screening LTFU.  As stated above, we do not believe that LTFU needs to be managed just in public health newborn screening programs. In some states it may be better managed in partnerships between states.

Minor Points

  1. Ref 16 has a dead link. I think the correct link is https://www.hrsa.gov/sites/default/files/hrsa/advisory-committees/heritable-disorders/meetings/day2-kus-follow-treatment.pdf Accessed 3/15/24

Thank you. We have actually gotten rid of this reference.

  1. This is a short slide set that does not seem to include the “data elements” referenced in the manuscript. The slides only propose that minimal data elements be considered. There may be a better reference from ACHDNC FUTR.

The reviewer is correct, and we agree with the assessment by reviewer one that we have likely given too much weight to this slide set.  The elements were discussed, but never voted upon by the ACHDNC, and have not appeared in peer reviewed literature. The awardees were discussing the same elements, even in the absence of these elements discussed at this meeting.  We have therefore removed reference to this brief slide set.

We do, however, want to give credit to the group who originally identified what we refer to as the core data set. We do say in our write up and in figure 1 that the NBS LTFU Core Data Set our group started with was suggested by in 2016 by a workgroup of NBS professionals organized by the National Coordinating Center (NCC) of the Regional Genetics Network and the Newborn Screening Translational Research Network [NBSTRN]. There was no formal write up, but we do cite personal communication with a member of NBSTRN who was instrumental in the meeting.

  1. Sometime in the future it may be possible to pair NBS records with death certificates to get a better idea of mortality due to screened diseases. The task is not trivial.

We fully agree.

  1. “Appropriate care” seems to be determined locally. Some disorders have consensus guidelines from professional organizations and those should be used when available.

We fully agree, and each program aimed to apply consensus guidelines when available. If these standards are adopted more broadly, we would encourage an adoption of consensus guidelines across all LTFU programs as they apply these data elements.

  1. I do not understand why UCSF and ACMG consider themselves NBS programs. I think the manuscript states that each of them has only a narrow focus of diseases represented in the data.

UCSF and ACMG do not consider themselves to be NBS programs. This is a report of six awardees funded under the same program working together to define a minimal LTFU data set for NBS. Being a NBS program was not a requirement for funding, nor is there any expectation that LTFU programs need to be based in NBS programs. The home for LTFU may be best placed in clinical facilities, with research entities, or with the public health NBS programs.

Please note on lines 137 to 140: “The purpose of the program is to support comprehensive models of long-term follow-up that demonstrate collaborations between clinicians, public health agencies, and families” (HRSA-21-079 funding opportunity announcement, pg. i). This was meant to be a demonstration project and to create models of LTFU and some groups chose to focus on a single condition or group of conditions to develop the model. None of the programs include all of the RUSP conditions due to the scope of their awarded proposals.  

  1. The yellow oval in Figure 1 should include NBS programs as part of “State Public Health”.

Thank you so much for the suggestion. That change has been made.

Reviewer 3 Report

Comments and Suggestions for Authors

This paper tackles an important implementation question of how it works, in practice, to gather data towards a minimal long-term follow up dataset for newborn screening programs. 

Unfortunately the methods and results are disorganized and limit the usefulness of the findings. I have included some detailed comments below.

- recommend inclusion of a table that outlines the ACHDNC minimal dataset variables and the revised set used by the authors. 

- It might be useful to use the term "revised minimal LTFU data set", since what the authors are proposing is a revision to the initial minimal LTFU data set proposed by ACHDNC.  

- Results section 3.1 and 3.1.2 seem to repeat the same information

- in some places, the authors distinguish children who moved and died, but in other places these numbers are combined. I understand combining these for the purposes of a denominator (i.e., need to know the number of living, diagnosed children within a jurisdiction), but otherwise separating out the number of children who died from those who were lost to follow up because they moved is very important.

- It is unclear how the ACMG program data contributed. Are these data from across the US? Could there have been overlap with cases from the other programs? These seem different enough from the other data included that their inclusion is questionable

- In section 3.2 it states "not all programs were able to report for all birth cohorts". This suggests that some cells in Table 3 contain data from only some of the programs?  If true, this makes it very difficult to interpret the results.

- If the authors can comment on any observed differences in rates of follow-up across the programs, that would strengthen the paper. This is mentioned briefly in 4.1 but no data is shown that actually compares follow-up rates across programs

Comments on the Quality of English Language

This paper could use a close edit. There are extra/missing words throughout, and some points are obscured by unclear language. In addition, the formatting of the paper is very messy, with inconsistent fonts throughout. 

Author Response

Dear Reviewer Three:

First off, thank you so much for taking the time and providing such thoughtful comments and suggestions. It is always helpful to see where our ideas are lacking clarity. Your feedback has pushed us to be more specific and has made this paper stronger.

One key area that this review helped us to solidify was where the data was extracted from and who the responsible parties for LTFU are. The collaboration across the six programs highlights that the programs can be managed by different types of entities, yet these data elements can successfully be used by all of the entities to assess LTFU.  We have tried to reframe our write up to reflect that.

In addition, we knew when we wrote this that there was never an adoption by the ACHDNC or even the ACHDNC FUTR workgroup of the core data elements. As you saw from our previous version, the only evidence we had was a slide (slide #4) from the workgroup report out at the April 24, 2019, ACHDNC meeting. Two of our authors were part of this work group. As we worked on this revision, there was a realization that the questions came from a workgroup organized by the National Coordinating Center (NCC) and NBSTRN. Again, there was no formal write up, but we do cite personal communication with a member of NBSTRN who was instrumental in the meeting. We now state in the paper that the NBS LTFU Core Data Set our group started with was suggested in 2016 by a workgroup of NBS professionals organized by the National Coordinating Center (NCC) of the Regional Genetics Network and the Newborn Screening Translational Research Network (NBSTRN).

Another major change, as you suggested, was to shorten the literature review and tighten up sections 3.1 and 3.1.2. We have made those changes and hope that we have still provided enough context for the readers.

Finally, we realized during the re-write that we needed to be specific that we were speaking about US newborn screening and have tried to clarify that as well.

We address each specific recommendation below. 

Thank you again for taking the time to review this article.

Reviewer 3

This paper tackles an important implementation question of how it works, in practice, to gather data towards a minimal long-term follow up dataset for newborn screening programs. 

Unfortunately the methods and results are disorganized and limit the usefulness of the findings. I have included some detailed comments below.

We agree with the assessment by reviewer one that we have likely given too much weight to this slide set.  The elements were discussed, but never voted upon by the ACHDNC, and have not appeared in peer reviewed literature. The awardees were discussing the same elements, even in the absence of these elements discussed at this meeting.  We have therefore removed reference to this brief slide set.

  1. It might be useful to use the term "revised minimal LTFU data set", since what the authors are proposing is a revision to the initial minimal LTFU data set proposed by ACHDNC.

We did not change to “revised minimal LTFU data set”. In replying to these reviews, we realized that ACHDNC never suggested or recommended these LTFU data elements. Instead, they were part of a discussion in the ACHDNC LTFU workgroup. Two of our authors were part of that group.

We do, however, want to give credit to the group who originally identified what we refer to as the core data set. We say in our write up and in figure 1 that the NBS LTFU Core Data Set our group started with was suggested by in 2016 by a workgroup of NBS professionals organized by the National Coordinating Center (NCC) of the Regional Genetics Network and the Newborn Screening Translational Research Network (NBSTRN). There was no formal write up, but we do cite personal communication with a member of NBSTRN who was instrumental in the meeting.

  1. Results section 3.1 and 3.1.2 seem to repeat the same information

Thank you. We fully agree with this. We have edited Sections 3.1 and 3.12 to reduce the repetition.

  1. in some places, the authors distinguish children who moved and died, but in other places these numbers are combined. I understand combining these for the purposes of a denominator (i.e., need to know the number of living, diagnosed children within a jurisdiction), but otherwise separating out the number of children who died from those who were lost to follow up because they moved is very important.

Thank you for this comment. The intention of our manuscript and the data elements described is to assess one year of follow-up in a state. To accurately calculate the denominator for the calculations of individuals in long-term follow-up care for 2022 we first removed the children who were not eligible from care due to moving out of state or being deceased.  We are not categorizing the children who moved as lost to follow-up, but rather not eligible for to be in the denominator as eligible for follow-up care in that year.

The ACMG data are indeed collected from across the US. The ACMG model is a different model of LTFU, and we have demonstrated that all the models presented here are capable of using this tool for assessing LTFU in their programs. We did not enter this activity with the assumption that all LTFU programs needed to be managed in state based NBS programs. We have edited the manuscript to make that clearer.

  1. In section 3.2 it states "not all programs were able to report for all birth cohorts". This suggests that some cells in Table 3 contain data from only some of the programs? If true, this makes it very difficult to interpret the results.

The data in Table 3 are representative of the goals and scope of the programs and are presented with the goal of demonstrating the ability of these data elements to measure LTFU across multiple programs. Comparisons between cohorts and between disorders is not intended. We have edited the table caption to help demonstrate our intent.

Thank you for this note. We agree that this would be interesting to point out, but this is not the intent of the manuscript, nor were data collected in a way to accurately make comparisons. Our intent is to demonstrate the use of a tool that could be used across different programs that are managed by clinical staff, research programs, or NBS staff. We believe that comparisons about the success of NBS LTFU systems should be reserved for future research that applies this tool among larger numbers of programs. Additional inference at this level could lead to spurious conclusions due to small numbers.

Comments on the Quality of English Language

  1. This paper could use a close edit. There are extra/missing words throughout, and some points are obscured by unclear language. In addition, the formatting of the paper is very messy, with inconsistent fonts throughout. 

We do not see any font changes in this version but will pay close attention when we convert this to a PDF. Thank you for reminding us that we need to do this.

Round 2

Reviewer 1 Report

Comments and Suggestions for Authors

The authors were fully responsive. The revised manuscript is very clear and informative. 

Author Response

Thank you for your review. We are glad we were able to answer your questions and address your concerns.

Reviewer 2 Report

Comments and Suggestions for Authors

This is a good start towards systematizing long-term follow-up.

Author Response

Thank you so much for your review.  We appreciate you taking the time to provide such thorough feedback.  As we stated last time, your comments enabled us to submit a stronger paper.

Reviewer 3 Report

Comments and Suggestions for Authors

The manuscript is greatly improved from the previous version. I recommend a few small additions that will strengthen its relevance for other NBS programs. 

- I recommend including a table that defines the final 5 core data elements that were used in the paper's analysis, for quick reference by readers

- can the authors elaborate on the barriers to gathering all 9 initial data elements? (results section 3.1)

- can the authors clarify where they derived the recommended cadence of care that was used to determine the "appropriate care" data element?

- in the discussion, can the authors elaborate on which data elements they would next like to see added to this core minimum set?  Thoughts on which additional elements would be most feasible to collect and most important for tracking LTFU program success would be instructive. 

Comments on the Quality of English Language

The language is improved but some copy editing is still needed in places. For example, at the top of p 13 the sentence: "Appropriate care was defined as the number of children seeing the appropriate specialist on the recommended cadence for care within their state/jurisdiction for the final data elements". The final five words (for the final data elements) should be edited out. 

Author Response

Dear Reviewer Three:

First off, thank you again for looking over our manuscript and providing useful suggestions. We worked to address each of your recommendations.

  1. You recommend including a table that defines the final 5 core data elements that were used in the paper's analysis, for quick reference by readers.

We included this table, labeled table 2.  Under section 3.1

Variable

Definition

Number of Children Diagnosed as having the condition

·     The condition was screened for using NBS dried blood spot.

·     The NBS was either a true positive of a false negative.

·     The child was confirmed to have the condition following the NBS.

Number of Children Who Died or Moved Before the Year Reported

·     The child died before January 1st for the year being reported.

·     The child moved out of the jurisdiction of the program (e.g. NBS, clinic) providing the LTFU data before January 1st of the year being reported.

Number of Children with Condition Who Are Alive

·   Child is not classified as deceased in the EHR; or

·   Child is not classified as deceased in the state vistal record system; or

·   Child is not reported as deceased to the LTFU program.

Number of Children Who Had Contact with a Specialist for Their Disorder within the Last 12 months

·   There is record that the child/family met with the specialist in person, via telehealth, or on a phone call; or

·   There is record that the child/family communicated with the specialist using email.

Number of Children Who Received Appropriate Care Specific to the Diagnosis within the Last 12 month

·   did the child see the appropriate specialist on the recommended cadence for care within the state/jurisdiction

  1. You asked us to elaborate on the barriers to gathering all 9 initial data elements? (results section 3.1)

Under section 3.1 we have now written: It quickly became clear that gathering all nine data elements was difficult for a few reasons. First, because most programs relied on EHRs or data from specialty clinics, it was not possible to know if every child had seen a PCP.  It was also difficult for some of the programs to access the NewSTEPs data; a few who could access this data noted it was difficult to resolve missing data because the NewSTEPs data does not have identifiers. The group also found that it was difficult to determine if the cause of death was related to the genetic condition because of the many ways the cause of death can be recorded. The group agreed to two data elements to create a denominator and three data elements to measure LTFU outcomes that were potential indicators of NBS success (See Table 2).

  1. You asked us to clarify where we derived the recommended cadence of care that was used to determine the "appropriate care" data element?

We included the following explanation in table 2. The definitions of appropriate care were derived using a crosswalk between the recommended visit cadence from the participating programs, which were informed by practitioners and care guidelines at the jurisdictional level. 

  1. You requested that we elaborate on which data elements we would next like to see added to this core minimum set in our discussion.  

We added the following to the end of section 4, just prior to section 4.1.

“Once these LTFU data elements have been tested and refined so a majority of LTFU programs in the US can provide data, it would be helpful to identify other data elements that intersect with public health, research, and clinical care (see Figure 1) that can be gathered consistently, reliably, and with minimal effort. When looking at figure one, a potential place to start the conversation might be around healthcare utilization, including access to a medical home.”

  1. You also noted that “the language is improved but some copy editing is still needed in places. For example, at the top of p 13 the sentence: "Appropriate care was defined as the number of children seeing the appropriate specialist on the recommended cadence for care within their state/jurisdiction for the final data elements". The final five words (for the final data elements) should be edited out.”

Thank you we made this edit and re-reviewed our article . We believe we have caught all the needed edits. Thank you again for you careful review.
